# Differential Response of Mycosis Fungoides Cells to Vorinostat

**DOI:** 10.3390/ijms24098075

**Published:** 2023-04-29

**Authors:** Zachary A. Bordeaux, Sriya V. Reddy, Kevin Lee, Weiying Lu, Justin Choi, Meghan Miller, Callie Roberts, Anthony Pollizzi, Shawn G. Kwatra, Madan M. Kwatra

**Affiliations:** 1Department of Dermatology, Johns Hopkins University School of Medicine, Baltimore, MD 21205, USA; 2Department of Anesthesiology, Duke University School of Medicine, Durham, NC 27710, USAmadan.kwatra@duke.edu (M.M.K.); 3Department of Oncology, Johns Hopkins University School of Medicine, Baltimore, MD 21205, USA; 4Department of Pharmacology and Cancer Biology, Duke University School of Medicine, Durham, NC 27710, USA

**Keywords:** mycosis fungoides (MF), cutaneous T-cell lymphoma (CTCL), histone deacetylase inhibitor, vorinostat

## Abstract

Mycosis fungoides (MF) is the most common form of cutaneous T-cell lymphoma (CTCL) and is characterized by epidermotrophism of malignant CD4+ T-lymphocytes. When MF advances to a recurrent stage, patients require treatment with systemic therapies such as vorinostat, a histone deacetylase inhibitor. While vorinostat has been shown to exhibit anti-tumor activity in MF, its exact molecular mechanism has yet to be fully discerned. In the present study, we examined the transcriptomic and proteomic profiles of vorinostat treatment in two MF cell lines, Myla 2059 and HH. We find that vorinostat downregulates CTLA-4, CXCR4, and CCR7 in both cell lines, but its effect on several key pathways differs between the two MF cell lines. For example, vorinostat upregulates CCL5, CCR5, and CXCL10 expression in Myla cells but downregulates CCL5 and CXCL10 expression in HH cells. Furthermore, vorinostat upregulates IFN-γ and IL-23 signaling and downregulates IL-6, IL-7, and IL-15 signaling in Myla cells but does not affect these pathways in HH cells. Although Myla and HH represent established MF cell lines, their distinct tumor origin from separate patients demonstrates that inherent phenotypic variations within the disease persist, underscoring the importance of using a variety of MF cells in the preclinical development of MF therapeutics.

## 1. Introduction

Mycosis fungoides (MF) is the most common form of cutaneous T-cell lymphoma (CTCL), accounting for approximately half of all primary cutaneous lymphoma cases [1,2,3]. MF typically follows a protracted course, with patients slowly progressing through patch, plaque, and tumor stages over several years [1,3,4]. However, these patients often suffer from a significant decrease in quality of life secondary to intense pruritus, sleep disturbances, psychosocial dysfunction, and a high cost of healthcare utilization [5,6,7,8,9]. Furthermore, MF is difficult to treat, with patients only rarely achieving long-term remission or complete cure, especially in advanced stages, for which the survival rate may be as low as 18% [1,10].

In the early stages, treatment of MF is focused on controlling cutaneous lesions by using skin-directed therapies such as topical corticosteroids, topical mechlorethamine, topical retinoids, ultraviolet light therapy, and radiation therapy [1,10,11,12]. However, in the later stages, treatment relies on systemic agents such as interferons, retinoids, chemotherapeutics, and monoclonal antibodies against common T-cell receptors [1,10,11,12]. One such systemic medication, vorinostat, is a class I and II histone deacetylase inhibitor that is approved for patients with persistent, progressive, or recurrent CTCL after the failure of two other systemic therapies [1]. Vorinostat is thought to induce the accumulation of acetylated histone and non-histone proteins, which results in chromatin remodeling, increased expression of tumor suppressor genes, cell cycle arrest, and apoptosis [13]. In MF specifically, vorinostat exerts anti-tumor activities by interfering with T-cell receptor signaling, suppressing the MAPK pathway, and downregulating anti-apoptotic genes such as c-FLIP [14,15,16]. However, the exact molecular mechanisms of vorinostat in MF remain unknown [13].

This study was undertaken to further characterize the effects of vorinostat in CTCL by using two MF cell lines, Myla 2059 and HH. We find that vorinostat downregulates CTLA-4, CXCR4, and CCR7 in both cell lines but differentially modulates other chemokine receptors and inflammatory pathways.

## 2. Results

### 2.1. Transcriptomic and Proteomic Effects of Vorinostat Treatment

The overall study design is shown in Figure 1. We first analyzed control and vorinostat-treated Myla 2059 and HH cells by using RNA sequencing to identify the targets that are affected by the medication. Differential expression analysis identified 5862 differentially expressed genes (DEGs) between control and vorinostat-treated Myla cells (Figure 2A) and 6277 DEGs between control and vorinostat-treated HH cells (Figure 2B). Gene set enrichment analysis (GSEA) based on the Biocarta database revealed the suppression of several inflammatory pathways which differed between the two cell lines. Treated Myla cells showed suppression of IL-12 and IL-17 signaling in addition to the downregulation of TCR, MHC, T helper cell, and cytotoxic T cell-related pathways (Figure 2C). In contrast, treated HH cells showed suppression of the IL-2RB pathway (Figure 2D). Notably, CTLA-4 signaling was suppressed in both cell lines following vorinostat treatment.

To broaden our understanding of the targets affected by vorinostat, we next subjected control and treated cells to RPPA. Functional enrichment analyses were performed on gene lists of proteins that are downregulated by vorinostat by using EnrichR. We first identified the top 10 pathways suppressed by vorinostat and then sought to confirm the suppression of the CTLA-4 pathway at a proteomic level. We found that the CTLA-4 pathway was indeed suppressed in both vorinostat-treated Myla (Figure 2E) and HH cells (Figure 2F). Additionally, the analysis revealed suppression of several growth-promoting pathways in vorinostat-treated Myla cells, such as MAPK, eIF4e, HER2, PI3K, and mTOR signaling (Figure 2E). Treated HH cells also showed suppression of mTOR and HER2 pathways, as well as IL-2RB signaling, similar to the transcriptomic analysis findings (Figure 2F).

### 2.2. Vorinostat Modulates the Expression of Chemokine Receptors and Ligands

Given the prominent downregulation of several immunologic pathways by vorinostat, we theorized that the medication may modulate the tumor microenvironment by altering the expression of chemokine receptors and ligands. We first identified the basal expression level of these receptors and ligands in Myla and HH cells by using RNA sequencing. As can be seen in Figure 3A, both cell lines showed prominent expression of *CCR4*, *CCR7*, *CXCR4*, and *CXCL10*. However, expression of other chemokines differed between the two MF cell lines. Myla cells expressed several receptors and ligands that were absent in HH cells, including *CCL1*, *CCR3*, and *CXCL3*, while HH cells expressed *CCL28* and *CXCR3*, which were absent in Myla. We next evaluated how the expression of these receptors changed following vorinostat treatment. Heatmaps of significantly different chemokine receptors and ligands between the control and treated Myla and HH cells are shown in Figure 3B and Figure 3C, respectively. In both cell lines, vorinostat suppressed the expression of *CXCR4* and *CCR7*, while it upregulated *CXCL11*. However, vorinostat affected several other receptors and ligands in these cell lines differently. *CCL5* and its receptor *CCR5* were upregulated by vorinostat in Myla cells, while *CCL5* was suppressed in vorinostat-treated HH cells. Similarly, vorinostat treatment upregulated *CXCL10* in Myla cells but downregulated its expression in HH cells. In contrast, *CCR10* was downregulated in Myla cells but upregulated in HH cells in response to vorinostat treatment.

We next sought to confirm alterations of some chemokine receptors by utilizing flow cytometry and RPPA. Consistent with our transcriptomic data findings, flow cytometry analysis also showed that the expression of CXCR4 significantly decreased in both cell lines following vorinostat treatment (Figure 3D). Additionally, the expression of CCR5 significantly increased in treated Myla cells, while there was no significant difference in treated HH cells (Figure 3E). We next conducted an EnrichR enrichment analysis on gene lists of proteins that were downregulated by vorinostat in both cell lines to determine if CXCR4 and downstream signaling pathways were suppressed at a protein level. This analysis showed suppression of Akt, ERK1/ERK2, Ras, Rac1, and CXCR4 signaling pathways in both cell lines following vorinostat treatment (Figure 3F,G).

### 2.3. Vorinostat Modulates Cytokine Signaling

Finally, given the suppression of various inflammatory pathways identified via GSEA, we proceeded to examine how vorinostat affected additional cytokine signaling pathways by using gene set variation analysis (GSVA). Boxplots of GSVA results are shown in Figure 4A–I. Interestingly, similar to the GSEA findings (Figure 2C,D), vorinostat had a differential effect on several cytokine pathways in the two cell lines and seemingly provided broader immunoregulatory effects on Myla cells. IFN-γ and IL-23 signaling were increased in vorinostat-treated Myla cells, while IL-2, IL-3, IL-4, IL-6, IL-7, and IL-15 signaling pathways were decreased. No significant difference was observed for these pathways in treated HH cells. Similarly, IL-5 signaling was suppressed in treated HH cells, but no difference was observed in treated Myla cells.

## 3. Discussion

The present study examines the transcriptomic and proteomic impacts of vorinostat in two MF cell lines. Our results demonstrate that vorinostat suppresses CTLA-4 signaling and decreases the expression of *CXCR4* and *CCR7* in both cell lines. However, the effect of vorinostat on other chemokines and interleukin signaling pathways varies between the two cell lines.

The ability of vorinostat to suppress CTLA-4 in both MF cell lines is noteworthy. CTLA-4 is an immune checkpoint co-stimulatory protein that is commonly overexpressed in various solid and hematologic malignancies and serves to promote evasion of immune surveillance by inhibiting T-cell activation and proliferation [17,18]. CTLA-4 expression is often increased in patients with MF and becomes progressively more dysregulated with advancement to the late-stage disease [19]. This may be one reason that vorinostat is efficacious in treating advanced MF and other hematologic malignancies [20,21], as suppression of CTLA-4 likely promotes a more robust immune response against malignant tissues [22]. Consistent with previous research, our study also found vorinostat to repress the growth promoting the TCR signaling pathway [14]. However, we find that the medication downregulates this pathway in Myla cells but not HH.

Vorinostat also reduced levels of the chemokine receptors *CXCR4* and *CCR7* in both cell lines. CXCR4 is expressed by most hematopoietic cells. The receptor and its chemotactic ligand, CXCL12, function to regulate cell migration and are upregulated in various cancers, including MF [23]. Activation of the CXCR4-CXCL12 axis has been shown to promote tumor growth, angiogenesis, and metastasis in breast [24,25,26], colorectal [27], and lung cancers [28,29]. In CTCL specifically, CXCR4 plays a role in cell migration and chemotaxis and may promote cutaneous homing of cells through interactions between CXCR4 on malignant lymphocytes and CXCL12 on fibroblasts and dermal stromal cells [30,31]. This finding corroborates previous studies demonstrating *CXCR4* suppression in vorinostat-treated CTCL cells [14]. However, Wozniak et al. did not detect *CCR7* suppression by the medication. CCR7 is another homing chemokine receptor expressed by dendritic cells and T-cells that induces migration and invasion of malignant cells into lymph nodes through its ligand, CCL21, which is present on lymphatic vessels [32]. The expression of CCR7 has been shown to be a marker of advanced MF and has been correlated with subcutaneous involvement [33]. Taken together, this suggests that vorinostat may modulate the expression of these chemokine receptors and ligands to alter tumor cell-homing.

When the expression of other chemokine receptors and ligands is evaluated, however, vorinostat appears to differentially affect each cell line. For example, vorinostat increased the expression of *CCL5*, *CCR5*, *CXCL10*, and *CCR10* in Myla cells but decreased *CCL5* and *CXCL10*, upregulated *CCR10*, and had no significant effect on *CCR5* in HH cells. The role of the CCR5-CCL5 axis in CTCL is not completely understood, and CCR5 expression may be limited to a rarer type of CTCL, namely, subcutaneous panniculitis-like T-cell lymphoma [34,35], so further studies are needed to determine how these changes may affect CTCL pathogenesis. CCR10 and its ligand, CCL27, are preferentially expressed on the memory-like regulatory T-cells in the skin, which help maintain cutaneous homeostasis [36]. CCR10 has also been shown to be significantly increased in CTCL patients regardless of their disease stage [37]. Regulatory T-cells inhibit anti-tumor immunity and promote tumor development and progression, so suppression of CCR10 in vorinostat-treated Myla cells could indicate less Treg activity and a more robust anti-tumor response [38]. Interestingly, *CXCL10* has previously been shown to be suppressed by vorinostat treatment, but its expression was not found to be differentially modulated by the medication in a cell-line-specific manner. CXCL10 is a cytokine that promotes the recruitment of CXCR3+ cells, such as Th1 and CD8+ T-cells [39]. Thus, the upregulation of *CXCL10* by vorinostat in Myla cells could promote a more robust cellular response against malignant T-cells. In fact, intratumoral CXCL10 is shown to be a positive prognostic factor for response to immunotherapy in other cancers such as melanoma [39]. In addition to *CXCL10* suppression, previous studies have demonstrated that vorinostat decreases the expression of several chemokine receptors and ligands, including *CXCR3*, *CXCL13*, *CXCL16*, *CCL1*, *CCL22*, *CCL26*, *CCL28*, *CCRA*, and *CCR4*, while upregulating *CXCL9*, *CXCL11*, *CCL19*, *CCL20*, *CCL27*, *CX3CL1*, *CCR2*, and *CCR6* [14]. We also detected the suppression of *CCL22* and *CCR4* with upregulation of *CXCL11* in both cell lines. However, *CCR6* was upregulated only in Myla cells but not affected in HH.

The effect of vorinostat on cytokine signaling is cell-type-specific. For example, vorinostat increased IFN-γ and IL-23 signaling, while it suppressed IL-6, IL-7, IL-15, and IL-17 signaling pathways in Myla but not HH cells. Previous studies have shown that vorinostat suppresses IL-4 and IL-5 and increases the expression of IL-6, IL-15, and IL-23 but did not detect a differential effect on these pathways [14]. IFN-γ is a proinflammatory cytokine that is often decreased in MF [40]. IFN-γ has been shown to stimulate CD8+ T-cells to promote a robust cellular response against malignant cells. Furthermore, IFN-γ supplementation has previously been used to treat several cancers, including MF [41]. IL-23 was shown to be increased in both epidermal keratinocytes and dermal lymphocytes, with a lower prevalence shown in later stages of CTCL, indicating that abnormal levels of IL-23 may also play a role in CTCL disease progression [42]. IL-6 is an acute-phase reactant that is produced by macrophages in response to pathogen-associated molecular patterns [43]. Elevated levels of this cytokine have been identified in CTCL patients and show a negative correlation with prognosis, as it may cooperate with vascular endothelial growth factor and leukotriene alpha to induce endothelial cell proliferation and neovascularization [44]. IL-7 is a cytokine that is necessary for T-cell development in the thymus, where it activates STAT3 and STAT5 to promote maturation of these cells [45]. Aberrant IL-7 singling is believed to a play role in the early stages of CTCL, as activation of STAT3 and STAT5 provides a survival signal to malignant T-cells until the later stages, in which there is often cytokine-independent JAK1 and JAK3 signaling [46,47,48]. IL-15 is an inflammatory cytokine that functions to promote cell growth and proliferation and is thought to have both anti-tumor and tumor-promoting activities, with its net effect likely being dependent on the particular tissue environment. This cytokine is thought to play a complex role in the pathogenesis of CTCL and is predominantly expressed in the skin lesions of advanced-stage MF, contributing to the survival and proliferation of malignant T-cells [49]. A recent study that utilized transgenic mice that overexpress IL-15 showed that this cytokine induces CTCL-like-lesions that are comparable to CTCL in humans. The same study demonstrated that ZEB1, a transcriptional repressor of IL-15, is unable to bind to the IL-15 promotor region in the cells of CTCL patients due to hypermethylation in this region of the gene transcript, resulting in overexpression of this cytokine [50]. IL-17 family cytokines are produced by many cells, including mast cells, neutrophils, and Th17 cells, and promote an immune response to extracellular pathogens [51]. IL-17 is shown to be expressed by malignant T-cells in CTCL and becomes progressively more dysregulated with disease progression [52]. IL-17 is thought to contribute to CTCL by activating pathways such as NF-kB and MAPK as well as by promoting angiogenesis [53,54].

Although both Myla and HH represent established MF, these cell lines originate from different patients and are, thus, expected to display molecular heterogeneity. For example, immunophenotyping studies have demonstrated differences in cluster of differentiation (CD) marker expression in these cells with Myla cells expressing CD4 and CD8 and with HH expressing CD2, CD3, CD4, CD5, and CD30 but lacking CD8 and CD25 [55]. Likewise, clinical trials have demonstrated a differential response among patients treated with the drug, with only approximately 29.7% showing an objective response [56]. Given the findings of the present study, this differential response may be due in part to heterogeneity in the immunomodulatory effects of the drug among MF patients, further highlighting the utility of personalized medicine in this population. For example, vorinostat may be synergistic with a CTLA-4 inhibitor in patients with a similar phenotype to HH or Myla, as it would further antagonize this pathway. At the same time, biologics targeted at other cell surface markers may be more or less effective in combination with vorinostat, depending on the molecular characteristics of that patient’s disease.

The limitations of this study include the use of only two MF cell lines and the lack of in vivo experiments. Further investigation utilizing additional cell lines should be completed to gain more insight into the molecular mechanisms of vorinostat on more phenotypically different cell lines and to assess the replicability of our work. In conclusion, we identified heterogeneity in the response of MF cells to vorinostat, which highlights the importance of using a variety of MF cell lines in the preclinical development of novel therapeutics.

## 4. Materials and Methods

### 4.1. Cell Culture

Myla 2059 (University of Copenhagen, Copenhagen, Denmark) and HH (ATCC, Manassas, VA, USA; #CRL-2105) were the cell lines chosen for this study. Both represent immortalized patient-derived advanced-stage MF cell lines. Myla 2059 was derived from a skin plaque biopsy of an 82-year-old Caucasian male with Stage IIA MF. The HH cell line was derived from the peripheral blood of a 61-year-old Caucasian male with Stage IVB leukemic MF [55]. Cells were maintained in RPMI-1640 (Gibco, Billings, MO, USA; #11875093) supplemented with 10% fetal bovine serum and kept at a concentration of 1 × 10^5^ to 1 × 10^6^ cells per ml. For experiments, only cells with less than five passages were used.

### 4.2. mRNA Sequencing

Myla 2059 and HH cells were treated with vorinostat (5 µM in 0.5% DMSO) or vehicle for 4 h. Total RNA was isolated from cell pellets by using an RNAeasy plus kit (Qiagen, West Caldwell, NJ, USA; #74034) according to the manufacturer’s instructions. RNA-seq data were processed by using the TrimGalore toolkit version 0.6.5 (https://www.bioinformatics.babraham.ac.uk/projects/trim_galore accessed on 31 March 2023). Only reads that were 20nt or longer after trimming were kept for further analysis. Reads were mapped to the GRCh38v93 version of the human genome and transcriptome [57] by using the STAR RNA-seq alignment tool [58]. Reads were kept for subsequent analysis if they mapped to a single genomic location. Gene counts were compiled by using the HTSeq tool [59]. Only genes that had at least 10 reads in any given library were used in subsequent analyses. Normalization and differential expression were carried out by using the DESeq2 [60] Bioconductor [61] package with the R statistical programming environment version 4.0.2 (www.r-project.org accessed on 31 March 2023). The false discovery rate was calculated to control for multiple hypothesis testing. GSEA was performed by using the GSEA software version 4.3.2 [62,63], with the Biocarta database as reference. GSVA was conducted with the GSVA R Bioconductor package by using the R statistical programming environment [64]. DEGs were defined as coding genes with a log2-fold change >1 or <−1 and a false-discovery-rate-adjusted *p*-value < 0.05.

### 4.3. Reverse-Phase Protein Array (RPPA)

Myla and HH cells were treated with vorinostat (5 µM in 0.5% DMSO) or vehicle for 4 h. Cell pellets were isolated and incubated with lysis buffer (1% Triton X-100, 50 mM HEPES, pH 7.4, 150 mM NaCl, 1.5 mM MgCl_2_, 1 mM EGTA, 100 mM NaF, 10 mM Na pyrophosphate, 1 mM Na_3_VO_4_, 10% glycerol, protease inhibitor (Roche Applied Science, Penzberg, Germany; #05056489001) and phosphatase inhibitor (Roche Applied Science, Penzberg, Germany; #04906837001)) on ice for 30 min, then they were clarified via centrifugation at 10,000 r.p.m. After quantification of protein concentration, lysates were denatured with 4X sodium dodecyl sulfate (SDS) sample buffer (40% glycerol, 8% SDS, 0.25M Tris-HCl, pH 6.8, 10% (*v*/*v*) 2-mercaptoethanol) and boiled for 5 min. Samples were stored at −80 °C and then sent to MD Anderson’s RPPA core facility (Houston, TX, USA). Biocarta functional enrichment analyses were performed on gene lists of proteins for which the expression decreased with vorinostat treatment by using EnrichR [65,66,67].

### 4.4. Flow Cytometry

Viable 3–5 × 10^6^ cells were incubated with vorinostat (5 µM in 0.5% DMSO) or vehicle for 24 h at 37 °C in 5% CO_2_. Cells were collected after incubation and filtered through 40 mm cell filters to obtain single-cell suspensions. The cells were washed in RPMI-1640, washed with PBS, and stained for viability (Zombie Aqua Fixable Viability Kit, BioLegend, San Diego, CA, USA). Cells were then incubated with TruStain fcX (BioLegend, San Diego, CA, USA) to block Fc receptor binding and were resuspended for labelling with antibodies against extracellular markers. The cell surface markers were incubated with cells in Hanks Balanced Salt Solution with 2% Calf Serum, 5 mM sodium azide, and 5 mM 4-(2- hydroxyethyl)-1-piperazineethanesulfonic acid. The surface-labeled cells were fixed and permeabilized by using the BD Cytofix/Cytoperm buffer kit (BD Biosciences, Franklin Lakes, NJ, USA). The cells were further labeled for intracellular cytokine markers. The antibody-labeled cells were then washed in intracellular staining buffer and resuspended in stabilizing fixative (BD Biosciences, Franklin Lakes, NJ, USA). Cell acquisition was performed on the BD LSRFortessa flow cytometer (BD Biosciences, Franklin Lakes, NJ, USA), and data were analyzed by using Cytobank software (Cytobank, Mountain View, CA, USA). The absolute number of the corresponding cell population was calculated as the total number of live cells x % of the corresponding cell population/100.

### 4.5. Statistical Analysis

Data were analyzed by using GraphPad Prism 9.0 software (San Diego, CA, USA). Comparisons between groups were conducted by using Student’s *t*-tests. Differences with a *p*-value of <0.05 were considered statistically significant.

## Figures and Tables

**Figure 1 ijms-24-08075-f001:**
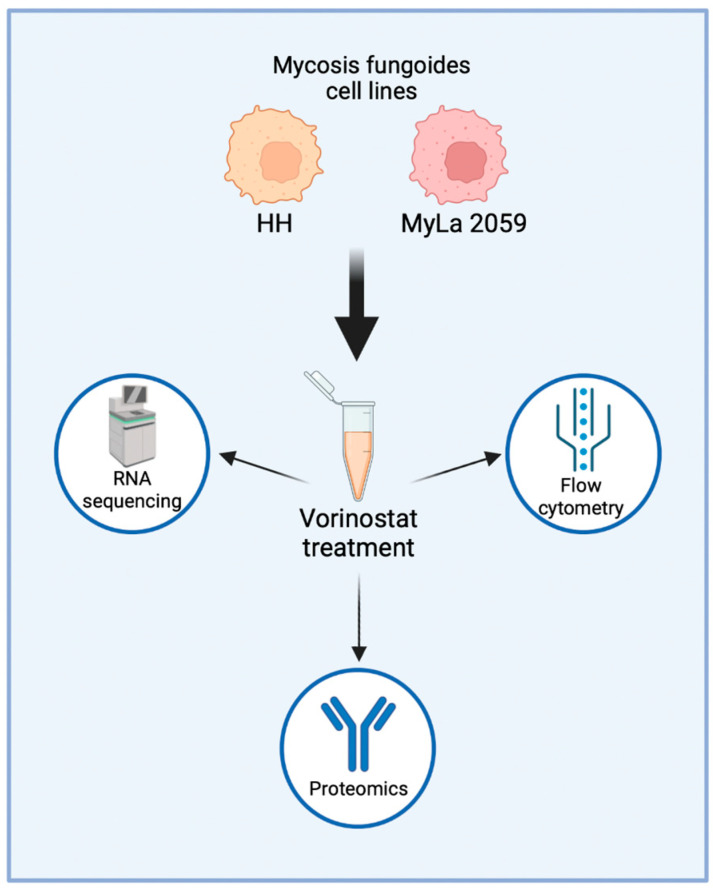
Overall study design. HH and Myla 2059 cells were used for in vitro experiments. Cells were treated with vorinostat or vehicle and subjected to RNA sequencing, reverse-phase protein array (RPPA) or proteomics, and flow cytometry.

**Figure 2 ijms-24-08075-f002:**
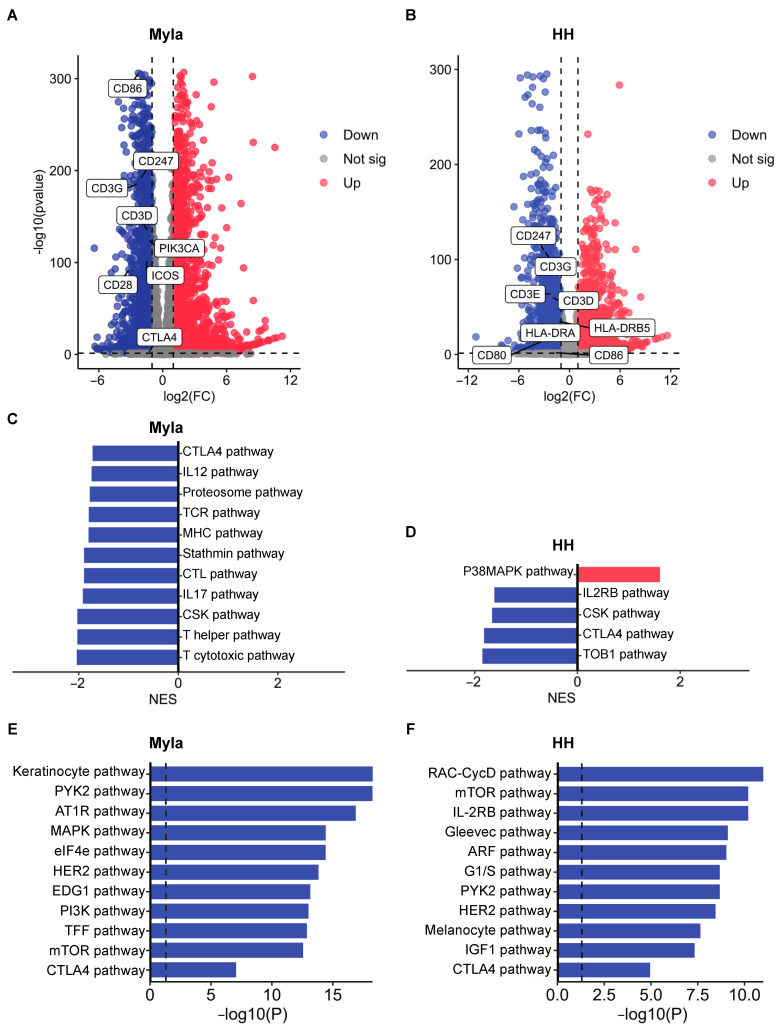
Transcriptomic and proteomics effects of vorinostat treatment. (**A**) Volcano plot showing DEGs between control and vorinostat-treated Myla cells. (**B**) Volcano plot showing DEGs between control and vorinostat-treated HH cells. Differentially expressed CTLA-4-signaling-related genes are highlighted in both volcano plots. (**C**) GSEA results showing pathways that are up- and downregulated by vorinostat in Myla cells. (**D**) GSEA results showing pathways that are up- and downregulated by vorinostat in HH cells. (**E**) EnrichR enrichment analysis of proteins found to be downregulated by vorinostat by RPPA in Myla cells. (**F**) EnrichR enrichment analysis of proteins found to be downregulated by vorinostat by RPPA in HH cells. FC—fold change; sig—significant; NES—normalized enrichment score.

**Figure 3 ijms-24-08075-f003:**
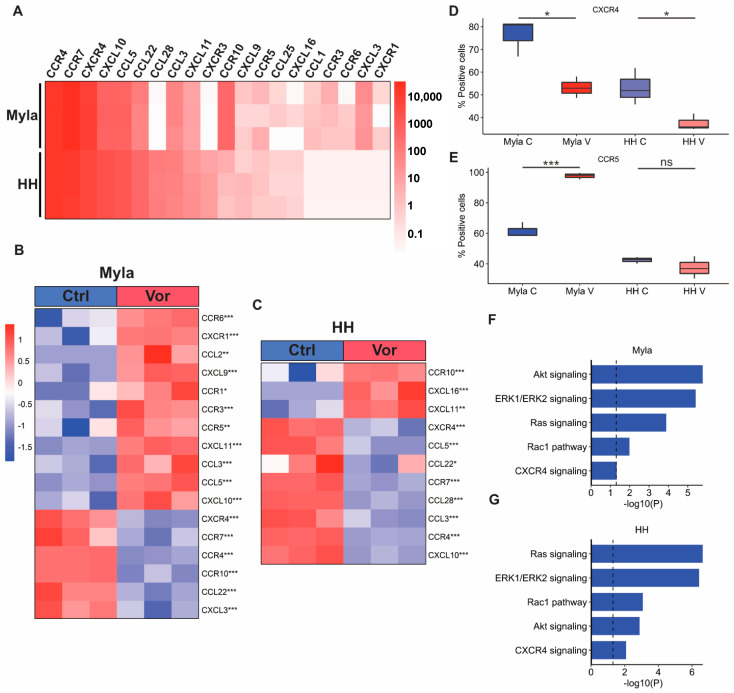
Effect of vorinostat on chemokine receptors and ligands. (**A**) Heatmap showing expression of chemokine receptors and ligands at baseline in Myla and HH cells. (**B**) Heatmap showing chemokine receptors and ligands significantly altered by vorinostat treatment in Myla cells. (**C**) Heatmap showing chemokine receptors and ligands significantly altered by vorinostat treatment in HH cells. (**D**) Boxplot showing CXCR4 expression in control and vorinostat-treated Myla and HH cells identified via flow cytometry. (**E**) Boxplot showing CCR5 expression in control and vorinostat-treated Myla and HH cells identified via flow cytometry. (**F**) EnrichR enrichment analysis of RPPA data showing downregulation of CXCR4-related pathways in Myla cells via vorinostat treatment. (**G**) EnrichR enrichment analysis of RPPA data showing downregulation of CXCR4-related pathways in HH cells via vorinostat treatment. * *p* < 0.05; ** *p* < 0.01; *** *p* < 0.001; Ctrl—control; Vor—vorinostat; C—control; V—vorinostat; NS—not significant.

**Figure 4 ijms-24-08075-f004:**
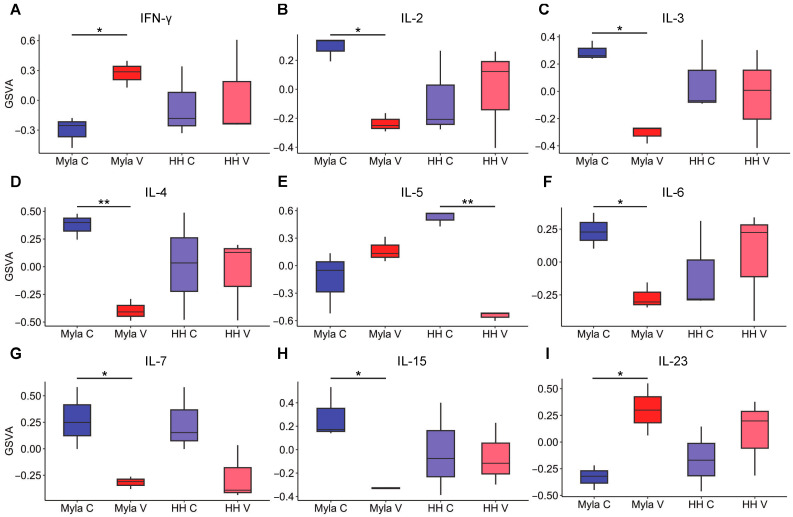
Effect of vorinostat on inflammatory pathways. (**A**) Boxplot showing effect of vorinostat on INF-γ signaling, (**B**) IL-2 signaling, (**C**) IL-3 signaling, (**D**) IL-4 signaling, (**E**) IL-5 signaling, (**F**) IL-6 signaling, (**G**) IL-7 signaling, (**H**) IL-15 signaling, (**I**) IL-23 signaling. * *p* < 0.05, ** *p* < 0.01; C—control; V—vorinostat; GSVA—gene set variation analysis.

## Data Availability

Available from the corresponding author S.G.K. upon reasonable request.

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
