# Peer review of "Differential Response of Mycosis Fungoides Cells to Vorinostat"

_ijms, 2023, doi:10.3390/ijms24098075_

Round 1

Reviewer 1 Report

This is an interesting and well written manuscript.  The study is well structured and conclusions are consistent with the experimental data.  The lack of in vivo studies is the main limitation and moreover the translational significance of the study should be better explained in particular with regard to the low efficace of the drug in the treatment of mycosis fungoides.

Reviewer 2 Report

I have no major comments for the authors. The study is designed and presented accurately. The authors also identify the limitations of the study. My major concern is relevance of the study. Vorinostat is currently used in treatment process for CTCL patients. In addition, effects of Vorinostat in CTCL cell lines has previously been studied not with RNA sequencing but with other methods. In my understanding, this study does not contribute any additional relevant information that has previously not been discovered. I would advice the authors to cite those previous studies and compare current data to signify the overall relevance of their discovery. 

None.
